# Growing Community: Factors of Inclusion for Refugee and Immigrant Urban Gardeners

Lissy Goralnik [1,*], Lucero Radonic [2] , Vanessa Garcia Polanco [3] and Angel Hammon [1]

1    Department of Community Sustainability, Michigan State University, East Lansing, MI 48824, USA
2    Department of Anthropology, Michigan State University, East Lansing, MI 48824, USA
3    National Young Farmers Coalition, Washington, DC 20002, USA
*    Correspondence: goralnik@msu.edu; Tel.: +1-517-353-5190

**Abstract:** Urban agriculture is an important neighborhood revitalization strategy in the U.S. Rust Belt, where deindustrialization has left blighted and vacant land in the urban core. Immigrants and refugees represent a growing and important stakeholder group in urban agriculture, including in community gardens across the Rust Belt Midwest. Community gardens provide a host of social and economic benefits to urban landscapes, including increased access to culturally appropriate food and medicinal plants for refugee and immigrant growers. Our work in Lansing, Michigan was part of a collaboration with the Greater Lansing Food Bank's Garden Project (GLFGP) to describe the refugee and immigrant community gardening experience in three urban gardens with high refugee and immigrant enrollment. Our research describes the ways garden management facilitates inclusion for refugee and immigrant gardeners and how particular factors of inclusion in turn contribute to social capital, an important outcome that plays a critical role in refugee and immigrant subjective wellbeing.

**Keywords:** social capital; community gardens; urban agriculture; Rust Belt Midwest; wellbeing; refugee and immigrant gardeners; garden management; agency; placemaking

## 1. Introduction

Urban agriculture, including community gardening, is an important neighborhood revitalization strategy in the U.S. Rust Belt, where deindustrialization has left blighted and vacant land in the urban core. Research on urban agriculture describes a number of associated social and economic benefits, including increased fruit and vegetable intake [1,2], beautification and community development [3], and job training [4]. When land and resources are available, urban agriculture can also make culturally appropriate food and medicinal plants more accessible for refugee and immigrant gardeners [5–7].

Refugees and immigrants are a growing stakeholder group in urban agriculture across the U.S. [8], and their experiences in the urban landscape are unique as new Americans. Many come from subsistence communities and integrate traditional ecological knowledge with the agricultural techniques and norms they are introduced to in the United States [5,9]. Others do not come with this rooted knowledge of food production, but discover in the process of gardening a connection to people, place, and the natural world [10]. Research supports the role of community gardening in strengthening connections to the garden and local communities for refugee and immigrant participants [11–13], or social capital, which, as Kingsley and Townsend explain [14], includes "[s]ocial networks, cohesion, support and connection facilitated by trust and reciprocity . . . [that] lead to material and social benefits such as social support and mobility" (p. 526).

Our work with refugee and immigrant gardeners in Lansing, Michigan was in collaboration with the Greater Lansing Food Bank's Garden Project (GLFGP). We conducted 31 interviews with 11 participants to investigate the refugee and immigrant community gardening experience in three urban gardens with high refugee and immigrant enrollment.

We were particularly interested in the ways gardeners felt included or excluded from the garden and urban communities and opportunities for the GLFGP network to better support their needs. This paper contributes to scholarship about the ways community gardening impacts social capital for refugee and immigrant growers in the Rust Belt Midwest and beyond.

## 2. Literature Review

### 2.1. Immigrants and Refugees in the US Food System

Over 13 percent of the U.S. population is foreign born [15]. In the United States, refugee status is a form of legal protection granted to people who have been persecuted or fear they will be persecuted based on race, religion, nationality, membership in a particular social group, or political opinion. In contrast, immigrants are often described as individuals who freely choose to migrate. However, as Holmes [16] explains, migration experiences are not a binary; rather they exist on a spectrum from forced to voluntary. We acknowledge that many individuals who may not qualify within the legal status of refugees left their home countries in response to economic, political, or social pressures. While immigrants and refugees may experience different journeys and have different legal status, they also share similar resettlement challenges related to employment, finances, language limitations, isolation, and acculturative stress [17].

Many immigrants and refugees in the U.S. find work in the industrial agriculture system as farm workers, in slaughterhouses, and in packing warehouses [18]. This dangerous labor entrenches inequities related to race, class, and education. In response to these inequities, scholars and practitioners have called for a return to more localized consumption practices, which, they argue, will foster deeper inclusion of traditionally underserved groups across the food system, including refugees and immigrants. [19–22]. However, this call for inclusion is nascent. While there is a rich interdisciplinary tradition that explores the connections between food, growing practices, and cultural identity, especially for racial and ethnic minorities in the U.S., immigrants and refugees have been largely absent from this narrative [18,19]. At the same time, these communities are an important and growing presence in alternative food systems, including urban agriculture [8].

### 2.2. Immigrants and Refugees in US Urban Agriculture and Community Gardens

Urban agriculture is a practice in or near an urban setting that includes backyard, roof-top and balcony gardening, community gardening and roadside urban fringe agriculture [23]. Long considered an alternative food space, urban agriculture provides a wide range of ecological, economic, and social benefits including promoting biodiversity [24,25] and building community [26]. These benefits have been posed as an opportunity to simultaneously address limited food access and vacant lots in the urban core in the Rust Belt Midwest [13,27,28], where deindustrialization in the early 21st century has led to poverty and population decline [29]. Recent literature highlights the important role of immigrant and refugee resettlement in revitalizing these spaces through participation in urban agriculture and community gardening initiatives [29].

Case studies of refugees and immigrants in community gardens highlight diverse motivations for and impacts of their involvement, including land tenure [30–32], reconnecting with agriculture [8], and community belonging [33]. Some research has shown that participation in community gardens can facilitate processes of inclusion to overcome cultural, social, and economic barriers [12], leading to increased social capital [34]. For example, a study of 55 community gardeners in Port Melbourne, Australia found that participation in the gardens connected participants with a community in which they had previously felt isolated [14]. However, there is also tension in the literature about the potential essentialism of claims that gardening can foster place relationships by reconnecting refugees and immigrants to subsistence practices [8]. Additionally, the literature suggests that some benefits of community gardening may not be equitably realized across communities because of the ways it contributes to gentrification, which can exclude marginalized groups, including

refugees and immigrants, from the very neighborhoods they have worked to develop [35]. However, these tensions are both incipient and contested [36]. Further exploration of the refugee and immigrant urban agriculture experience is necessary to understand how participation in community gardening impacts wellbeing.

*2.3. Garden Management*

While the objective and practice of community gardens in the United States has evolved over time, from Potato Patches in the Panic of 1893 to Victory gardens during World War II, the term 'community gardening' has gone unchanged [37]. However, over the last 15 years, the purpose of community gardening has shifted beyond food security to include urban planning and community development goals [38]. Neo and Chua [39] point out that the very definition of the 'community' in community gardening is often ambiguous, which has led to tension around who is included (or excluded) in garden spaces, and how. In the absence of clarity, the ways gardeners, managers, and funders conceptualize the purpose and function of a particular 'garden' determine who is served (or not served) by gardening activities, and garden-level management, including decision making about garden membership, leadership structures, and aesthetics, can impact participants' perceived inclusion in community garden spaces. Without attention to who is included in the garden experience, or to the nature of community in the garden space, urban agriculture can serve to re-entrench already existing inequities rather than contribute to neighborhood revitalization.

**Garden Membership**

Firth et al. [40] describe two primary types of garden membership: (1) required residency in certain geographic boundaries, i.e., place-based community, and (2) shared commitments, i.e., interest-based community, which might include commitments to environmental protection, food access, or health. These garden-level decisions about who gets to participate in garden activities determine participant relationships, experiences, and perceptions of inclusion. As such, they contribute to the development of social capital, i.e., "social resources inherent within community networks" (p. 38), including social cohesion and social interaction [41], which research shows play a critical role in refugee and immigrant subjective wellbeing [42]. Firth et al. [40] go on to describe four ways community gardens facilitate social capital: (1) bringing people together with a shared purpose around a common activity, which creates collective ownership, (2) providing a meeting place where people can interact and create community, (3) creating opportunities for informal interactions like growing, cooking, and eating food, which can bridge connections across communities, and (4) providing an opportunity to create connections with institutions and authorities. Shostak and Guscott [34] add to this list a fifth item, preserving cultural knowledges and practices, which is particularly important for refugee and immigrant gardeners.

**Garden Purpose and Leadership**

Research shows that inclusion in the community gardening space is affected by a garden's purpose, leadership structure, and acceptance within the wider community, each of which is impacted by network- and garden-level decision making. Neo and Chua [39] describe two primary purposes of community gardens: (1) "community-centric" and (2) "garden-centric". The former focuses on nurturing community within the context of gardening, while the latter prioritizes food production in the context of community. These central purposes, whether stated or implied, become reified by garden norms and expectations for participation. Although gardens can support both interests, generally one purpose assumes primacy and dictates a garden's culture. If gardeners' own priorities differ from the norms and implied purpose of the garden environment, then the gardeners may feel excluded from the space and experience, or feel that their contributions are not valued by the rest of the community.

Garden leadership structures also impact the participant experience. In a study of community gardens in Miami, Florida, Drake [4] identifies three garden leadership structures

with consequential impacts on community formation in the garden: (a) grassroots gardens, planned and managed by gardeners, (b) nonprofit-managed gardens, and (c) externally-organized gardens, developed or hosted by groups like academic extension or municipal agencies, who intend to hand control back to the community. In the grassroots gardens, where "gardening is a way for individuals to connect with others in their community, mobilize resources, and build social capital" (p. 179). Drake found that "inclusion revolved around whether or not people abided by rules of garden maintenance" (p. 187). In the other two garden types, garden success and inclusion instead hinged on the garden's ability to satisfy the goals of funders/organizers, which in these case studies included food security, food access, demonstration, and education. Grassroots gardens are driven by and depend upon stakeholder involvement at every level of garden management, and this buy-in, as well as the commitment to neighborhood wellbeing and the norms of garden leadership, Drake found, leads to inclusive, successful spaces. Facilitators of other types of gardens can learn from these outcomes to consider ways they might foster stakeholder involvement and voice within their own leadership structures.

**Garden Aesthetics**

Community perceptions of local gardens also contribute to the long-term success of gardens and the sense of inclusion for gardeners. Community perception is often determined by garden practices, which are dictated by the norms and expectations within particular gardens, and by cultural notions of beauty or function. For example, some gardeners prefer straight rows that are free of weeds, and this practice might be encouraged within some gardens; other gardeners find that mixed plants without borders suit their needs, and this technique might be reinforced by garden norms. Both types of gardens are rooted in different notions of what gardens "should" look like [13] and how they function. When the garden practices do not reflect the aesthetics of the wider community, though, there can be tension [25]. Pitt [43] notes that gardeners are at times fearful of unfamiliar plants or gardening methods, even when what appears to be chaotic or untended is an intentional strategy by other gardeners. Because "the 'untidy' look of some community gardens can challenge . . . ideas about the order of green spaces and gardening in the city," Strunk and Richardson [13] write, the desire for more organized spaces is often codified in local laws (p. 837), which can both situate immigrant and refugee gardeners outside local norms, and also make their gardening practices or traditional knowledge illegal or challenging to implement. Therefore, perceptions of inclusion from the beyond-gardening community are an important catalyst of perceived garden inclusion for gardeners.

## 3. Methods

### 3.1. Context and Study Site

Ten percent of the Lansing population is foreign born. Identified by the Brookings Institute as a top ten medium-sized metropolitan area in the United States for refugee resettlement [44], Lansing has welcomed more than 15,000 refugees from 48 countries since 1975, with 441 refugees settled in 2017 when the most recent data were collected [45], including many from Vietnam, Cuba, Iraq, Somalia, Burma, and Bhutan, among other places [46]. The city is also home to a vibrant urban agriculture community, including more than 100 community gardens in the greater Lansing area hosted by the GLFGP, which provides access to land, training and leadership workshops, tools, seeds, starts, and resources. All garden leaders in the network are community volunteers who either garden at the site and/or facilitate the garden for a site where they spend a lot of time, e.g., at a church. Therefore, while garden-level leadership is grassroots, these leaders are supported by top-down guidance and resources from the GLFGP. The GLFGP's primary purpose is to increase food security by promoting food production, access to culturally appropriate foods, and donation opportunities for local gardeners.

The GLFGP directly manages 18 of the 100 gardens in their network, and across those 18 gardens over 20% of participants self-identify as refugees or immigrants (personal communication, 2018). GLFGP reports that ten of the 100+ gardens are highly enrolled

by immigrant and refugee participants. Our research focused on three of these gardens (Table 1), identified by network leadership as sites that would be open to engagement with researchers. We also conducted observations at two other highly enrolled refugee and immigrant gardens and visited the remaining five. These are interest-based gardens (rather than place-based, i.e., gardeners are required to live in the immediate neighborhood). Gardeners join these garden communities by choice, usually because they are convenient and support refugee and immigrant participation.

**Table 1.** Study garden size, plot details, and refugee enrollment.

| Garden | Size (feet$^2$) | # Plots | Plot Size | % Refugee (in 2016; Immigration Status Not Tracked) |
|:---:|:---:|:---:|:---|:---|
| 1 | 12,269 | 22 | 325 ft$^2$ + 650 ft$^2$ | 6% (primarily international and immigrant enrollment) |
| 2 | 70,000 | 75 | Avg. 900 ft$^2$ | Est > 90% |
| 3 | 18,000 | 29 | Avg. 655 ft$^2$ | 10% (primarily international and immigrant enrollment) |

The lead author has been nurturing a collaborative relationship with GLFGP for over five years, which allowed the team to develop this project in response to the organization's expressed needs and interests. Several months prior to data collection, another author volunteered with the GLFGP's resource center to distribute seeds, tools, and information to gardeners for the season, volunteered during the garden leader training workshops, and took a plot at one of the study gardens, where she spent a summer growing food, observing garden dynamics, acquainting herself with the landscape, and celebrating the season as a member of the community. She conducted observations and interviews in the three study gardens late in the summer. This multi-level community investment provided the opportunity to develop organizational relationships and observe organizational practices.

### 3.2. Data Collection

In the summer of 2019, we used a convenience sampling protocol to recruit self-selecting gardeners from three GLFGP gardens serving primarily immigrant and refugee gardeners and snowball sampling to increase our sample pool [47]. Our final sample included 11 gardeners from Burma, Bhutan, Congo, Haiti, Kenya, Malawi, and Nepal. Table 2 shows the self-reported demographics of each gardener. Participant tenure in the US ranged from 5–20 years; most had been living in the US for 6–10 years (Table 2). Our sample size reflects best practices for case study research [48] and recent literature on refugees and immigrants in community gardens [49]. Participants were invited to engage in five separate data collection activities between June and September, including three short interviews (Appendix A), a drawing activity, and an image sorting activity. Here we present results from 31 interviews (three interviews per participant, except for one gardener from Garden 1 who only participated in the first interview). Collectively, the interviews focused on: (1) gardening practices, (2) motivations for participation, (3) sharing practices, and (4) sense of inclusion in the garden (Table S1). Questions were developed in conversation with the GLFGP community partners and guided by relevant studies [12,50–52]. Interviews were 15 min long on average. To acknowledge the participants' time, we offered a $10 grocery gift card for each of the five data collection events. We also offered translators, but all participants chose to participate in English. Interviews were audio recorded for accuracy and transcribed into text files. All protocol were approved by the GLFGP Refugee and Immigrant Liaison and deemed exempt by Michigan State University Institutional Review Board (study #00002500).

### 3.3. Data Analysis

Using an emergent thematic protocol [53], one author inductively coded 10 (of 31) interviews to create a working codebook, meeting weekly with another author to discuss observations and relationships across the themes as the codebook was developed. Similar

codes were grouped into themes (Table 3), labeled with participant language, and provided a narrative description in a spreadsheet. We then used this working codebook to analyze the rest of the data, adding new codes as they emerged, then adapting and streamlining the themes to reflect the additions. When no new codes emerged, we finalized the codebook and pulled exemplary text from the transcripts to create a data table [54]. We then used the final codebook to deductively re-code the complete data set three times.

**Table 2.** Sample population garden affiliation, origin countries, immigrant/refugee status, number of years in the US, number of years gardening in the community, and interview participation.

| Participant Garden | Country of Origin | Self-Identified Status | # Years in the US | # Years Community Gardening | Interview Participation | | |
| --- | --- | --- | --- | --- | --- | --- | --- |
| | | | | | #1 | #2 | #3 |
| 1 | Malawi | Immigrant | >16 | 4 | x | x | x |
| 1 | Malawi | Immigrant | >16 | 4 | x | x | x |
| 1 | Kenya | Immigrant | 11–15 | 9 | x | x | x |
| 1 | Malawi | Immigrant | 6–10 | 1 | x | | |
| 2 | Burma | Refugee | 6–10 | 8 | x | x | x |
| 2 | Bhutan | Refugee | 6–10 | 3 | x | x | x |
| 2 | Bhutan | Refugee | 6–10 | 4 | x | x | x |
| 2 | Bhutan | Refugee | 6–10 | 3 | x | x | x |
| 2 | Congo | Refugee | 3–5 | 2 | x | x | x |
| 3 | Kenya | Immigrant | 11–15 | 3 | x | x | x |
| 3 | Haiti | Refugee | >16 | 1 | x | x | x |

**Table 3.** Analytical categories and associated themes from the emergent thematic analysis. The themes are interpretive groupings of related codes identified during the analysis; categories reflect shared attributes of related themes.

| ANALYTICAL CATEGORIES | ASSOCIATED THEMES |
| --- | --- |
| Physical Space | Beauty |
| | Order and Organizational |
| | Access to resources |
| Agency | Learning and experimentation |
| | Being part of something larger |
| Personal Wellbeing | Traditional foods |
| | Health and safety |
| Community Wellbeing | Relationship building |
| | Being listened to and heard |
| | Sharing techniques and produce |

The primary coder wrote daily memos to reflect on the coding process, her experience in the field, and her observations across the data. Two other authors provided peer review during biweekly meetings to maintain consistency in the coding process [55]. Two colleagues un-involved in the study also provided peer review several times during the analysis.

## 4. Results

The interviews revealed ten primary themes that we grouped into four analytical categories (Table 3) that reflect their interactions and shared attributes: (1) Physical space, (2) Agency, (3) Personal Wellbeing, and (4) Community Wellbeing. Each of these categories describes the mechanisms that facilitate participants' sense of inclusion and belonging in the garden, or the ways that the physical and relational spaces of community gardening impacted their experience. We will describe each of these categories below, including examples from the interviews that capture the primary themes identified in the analysis.

### 4.1. Physical Space

Participants discussed gratitude for having a place to spend time outside, pride in observing progress in the garden, and joy in participating in the creation of beauty. These affective connections to the garden space were strengthened by physical investments in the site, e.g., placemaking, and by the natural assets of the site.

The access to space the garden provides is a benefit for participants, some of whom may have moved from more rural landscapes, do not have their own yards in the U.S., and/or live in tight quarters. One interviewee explained, "Because . . . you spend so much time inside here in Michigan, almost all the time is . . . inside in winter. Nowhere to go. This is the best way in summer to come out". In this way, the garden provides a sense of freedom. However, it is more than just an outdoor place to go. Several participants shared that they had previously grown plots in other community gardens and described particular characteristics of the physical environment that impacted their gardening experience. One participant discussed soil health, explaining that "Over there [at a different garden] the soil was sandy, too hot and didn't drain. And here [the soil] is better. Black". Other participants discussed the availability of water, straw, or mulch and the ways these resources contributed to their productivity and enjoyment in the garden. Participants' satisfaction with the physical space increased when they felt listened to by the GLFGP leadership about their resource needs: "If you're making a complaint... [the] administrators take it seriously. If we're complaining about something like the water . . . they did something . . . [W]hen you tell them something they respond . . . . [and] show that they care. And . . . that's what can make me comfortable. If they say we can't, we can't. But they listened, you know?" Being able to voice their resource needs and be heard, even when the desired resources could not be provided, was meaningful.

In addition to the natural assets of the physical space, participants discussed how particular factors at the site impacted their perceived safety. These features included the garden's geographic location in the city, the presence of fences, and garden lights. Discussing what they valued about their garden, one participant shared: "It's . . . just safe. I don't think somebody has complained about getting worried about being late, harvesting late, or staying there later, or coming and finding yourself [alone] in the garden." Feeling safe is an important part of a positive garden experience.

### 4.2. Agency

Participants appreciated the ability to exert agency in the physical landscape, or the feeling that they could contribute to the organization, beauty, and function of the place. One shared: "I think [choosing what to grow] is what I value the most" about the gardening process." The opportunity to exert choice in the garden provided a sense of empowerment, and for some participants, this choice also leads to deeper investment. "[W]hen you have a project," a participant shared, "you want to see it succeed. It start[s] when you are preparing your soil and planting, you start seeing the progress and you cherish that. So, being part of the garden I like it because you see what you grow and . . . harvest the impact of it." The garden experience provided an opportunity to participate in cycles, and the output of the initial labor was rewarding. Another explained, "I love when the garden is clean so whatever I can do to make it look better I will do it." Contributing to beautification and exerting autonomy to participate in a process from start to finish provides a sense of

ownership, which is enhanced when gardeners can play a role in developing the garden space to meet their needs. For example, one garden leader described a placemaking project in their garden: "We developed this play area, and . . . I just got a swing . . . for the adults. . . . [W]e have everything there [for] when the older adults are sitting in the shade, you'll find them visiting, or playing [a] game with each other . . . .[W]e want to make sure that everybody who comes to the garden feels welcome." Attention to the ways garden spaces can be used beyond the utility of food production signals that the space is a place for community building, which gardeners appreciated, and having the agency to shape the garden space and experience was important to participants.

Gardeners also discussed the opportunity to contribute to something larger than oneself by growing more than they need. For example, one participant explained, "[W]hen we plant our plants, we are also donating some to our community around us, so I feel like I am part of the community more. When [I] do that, I feel like I am contributing something." Another shared, "I grow something and then I donated to the soup kitchen and that makes me feel connected, [b]ecause I am doing something that can help someone." Gardeners appreciated the opportunity to share with their community because "I'm part of something positive," and "we know there [are] people in need of organic food and we are able to contribute that." A participant explained that they feel, "Pride after you've put in a lot of effort growing something and then you give it to somebody [and] they use it . . . You feel good helping somebody." The formal and informal participation of immigrants and refugees in produce sharing, knowledge exchange, and donation activities makes them assets in a community garden network that is rooted in addressing hunger, which in turn increases their sense of belonging or being valued in the system.

### 4.3. Personal Wellbeing

Participants described the ways their physical, emotional, and economic wellbeing were improved by community gardening. One participant explained that "gardening is my leisure time". Another shared that, "[The garden] offers me a form of exercise and provides some happiness . . . because when I see the plants growing, I just cherish that and I know I am getting something that is organic and so good for me". A number of participants shared that exercise and access to healthy or organic food were important drivers of their participation in the garden. Other participants found the garden experience important for their mental and emotional wellbeing. One participant shared that she "can talk to people [in the garden] after a long day of hard work so it is kind of therapeutic for me". Another found solace in the beauty of the landscape: "You get into the garden, you're depressed, then you come out like, 'Look at those flowers . . . They're so bright!'" The experience of being in the garden—through social interactions, nature contact, and engaging in activities they enjoy—contributed to broader wellbeing and health for the participants.

### 4.4. Community Wellbeing

A key benefit of the garden for several participants is the opportunity for co-learning and sharing, as one participant explained: "[B]eing part of the garden . . . gives me the opportunity to interact with my fellow gardeners and share what we have and also learn from my friends. So I find to be an enrichment opportunity." This sentiment was echoed by another participant: "The community purpose of the community garden is not just to plant plants and be done with it, but also to form some community bonding. So we learn from one another, we share experiences, and it is also a place where we can hang [out]." The chance to share knowledge and produce is a strong motivator that attracts many participants to the garden. Participants are proud to be able to provide for others, as well as grateful for the opportunity to receive when they are in need.

The sharing that occurs in the garden is not just a feature of being in the same space with others, but also a representation of community support. One participant explained: "[The garden] is like a home. Like a second home. . . . [Y]ou get everything from what you are growing and I feel like part of the community." One of the primary relationships

nurtured in the garden is between members of gardeners' own communities, e.g., Kenyan gardeners sharing with other Kenyans. A number of participants described distributing produce and sharing meals within their cultural community. One participant from Malawi explained, "I grow food that I wouldn't find in the store. All these vegetables here . . . like luni or chisaga." Another participant shared, "When I invite visitors [from] my same community I know I can cook something for them that they are familiar with. And they really appreciate it. . . . [T]hey come to my house, they will be like 'do you have that Kenyan vegetable over there?' Yeah, I got it. So, I know everyone eats that and they are happy." One gardener discussed growing kale and tomatoes for their friends, and in return "The ones that have been shared (sic) with me mostly have been managu [and] shitzaga, [which] . . . is pretty much spider flower." Collectively, then, gardeners support each other's dietary and cultural food preferences by sharing seeds, practices, produce, and recipes.

The gardens also offer opportunities for cross-cultural exchange beyond the garden. For example, one participant highlighted how she introduced a colleague to her own cultural foods: "Oh, it takes a while to develop the taste for it. I was surprised, I took some . . . to work the day before yesterday and let this girl taste it, she was like, 'Oh! I like it!' I'm like, 'Oh okay!' I will make some for you and bring it by." This sharing provides an opening for cultural recognition and appreciation and provides opportunities to identify shared flavors and experiences. As one participant from Haiti explained, "I have my neighbor . . . [she is] Dutch or something. And she loves coming out here . . . just to pick whatever... that seems familiar to her from her country . . . . I have stuff like that here [ . . . and] she enjoys that." This sharing outside one's immediate community, especially the recognition that one's cultural flavors also taste like home for someone from across the globe, is an important point of connection. Along similar lines, participants also detailed how cross-cultural exchanges extended to the realm of practices and techniques. For example, one participant shared:

> I can bring something [to the garden] that others don't know. Like [w]e have a special way of doing the weeding. So we can teach others that might do it differently. . . . And the different plants we plant. [T]he first time we planted amaranth, people were inquiring what is this and we share it the nutritional content of that and everything. I feel like [I am] able to contribute a little bit of knowledge there.

By contributing expertise, gardeners find an opening to build relationships and contribute to the wellbeing of the larger community.

Finally, the gardens also connect gardeners to the beyond-garden community. One participant shared: "[P]eople in the neighborhood, they say, 'Oh, hi ma'am, hi,' until you think you know them . . . .[T]hey know my face, which is a good thing, then you feel safe." Another participant shared: "I feel good because . . . [as] random people pass, you get to learn the environment in the garden and in the neighborhood . . . . [Y]ou see people every day until you start saying hi like you know them." As the participants described, their physical presence in the garden makes them visible to others and provides opportunities for interaction. It is good to be seen, and these casual conversations about gardening, crops, the weather, or just a simple exchange of pleasantries are important entry points into the beyond-garden urban landscape. Several participants expressed feeling a sense of social isolation before joining the garden, and interactions like these can help to transcend this feeling of being alone.

## 5. Discussion

Each of the categories discussed above, i.e., physical space, agency, personal wellbeing, and community wellbeing, meaningfully impacted participants' sense of inclusion in the garden. Figure 1 describes the relationships between these mechanisms of inclusion. Physical space and agency were the primary mechanisms; they catalyzed or facilitated the other two in a feedback dynamic, personal and community wellbeing, referred to here as ancillary mechanisms of inclusion. Both physical space and agency are directly impacted

by garden management decisions like membership structure, garden purpose, and garden leadership. Collectively, the four categories contribute to increased social capital, an important outcome for refugee and immigrant gardeners, who arrive having left behind familiar networks of social support and with few obvious avenues for mobility [56]. While many of our participants have lived in the US for quite some time (Table 2), "prior skills and training often do not easily transfer to the American labor force,... certifications, degrees, and licenses associated with ... professions are probably not recognized... [,and] families' previous social status and educational history do not provide advantages" (p. 948–949). These barriers are not easily transcended within a generation.

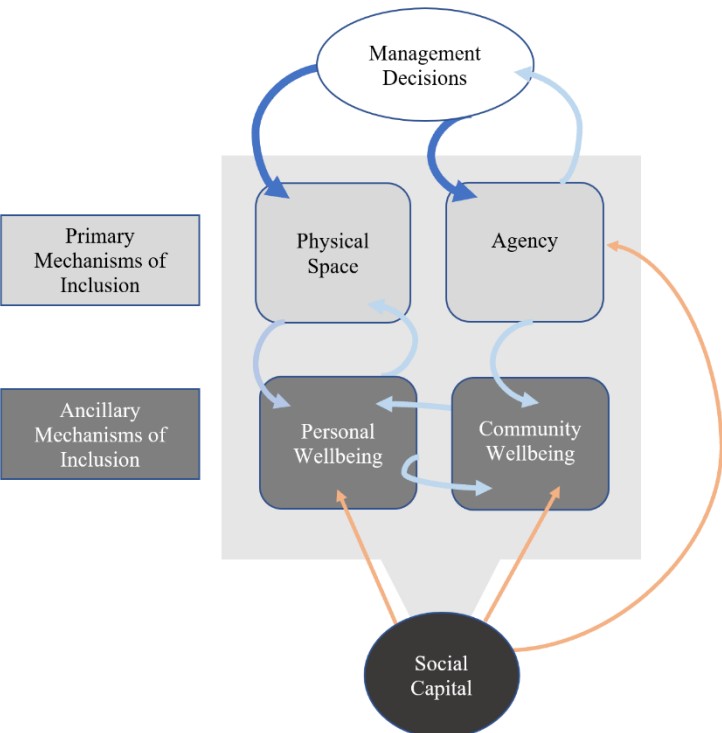

**Figure 1.** Conceptual framework detailing the interactions between garden management inputs, the analytical categories from our study, and the outcome of social capital. The dark blue arrows document the impacts of management decisions on mechanisms of inclusion. The light blue arrows indicate feedback relationships between gardener activities and the mechanisms of inclusion. The orange arrows indicate scholarly connections between social capital and the mechanisms of inclusion [34,42].

In the discussion we will explore the ways garden management impacts the identified mechanisms of inclusion, and how this inclusion impacts social capital. Our goal is to provide insight for other practitioners who aim to build inclusive garden spaces and to contribute to literature about the ways refugee and immigrant gardeners build social capital through the practice of community gardening. Our work is in conversation with literature on the relationship between community gardening and increased social capital, most directly with Firth et al.'s [40] typology of ways that community gardening leads to social capital and Shostak and Guscott's [34] addition of traditional knowledge to this list. Our contribution to this dialogue is a nuanced description of the relationships between garden management, perceived inclusion, and increased social capital for immigrant and refugee growers.

*5.1. Physical Space*

Access to the physical space of the gardens was directly dictated by management decisions about membership structure. The gardens in our study are interest-based, rather

than place-based. While they are conveniently located for many of the gardeners, enrollment is not geographically bound. In fact, several of the more seasoned participants in our study had previously gardened at other sites in the network, demonstrating they were willing to travel to garden and made choices about where to participate based on the experience and resources in particular gardens. The gardens in our study are known by GLFGP leadership as places with high refugee and immigrant enrollment, and as such they are provided regular access to the network Refugee and Immigrant Liaison. Gardeners in our study joined these gardens for this particular experience, which in turn increased their personal wellbeing.

Being able to use the physical space of the garden also increased wellbeing for participants in our study by providing access to fresh fruit and vegetables and reducing the cost of groceries. Scholarship shows that cost is a barrier to quality fresh and healthy food for immigrants and refugees [57]. While gardening provides a means to overcome this barrier, it also often requires labor and resource investments [58,59] that can offset the benefits of growing one's own produce. Because the GLFGP provides free access to many required inputs, though, including seeds, plants, tools, and education, our participants had both had an economic incentive to participate and were able to benefit from increased access to fresh produce. Additionally, the physical space of the garden impacted participant wellbeing by providing space to be outdoors, which in turn impacted physical and mental health. These are common benefits of community gardening for all groups [1], but particularly impactful for refugee and immigrant gardeners, many of whom live without ready access to outdoor spaces [60]. Our participants explained that because the garden made them feel good, they spent time there beautifying and enriching the social spaces of the garden. Gardener relationships with the physical space were circular in this way: the space provided a context for physical and mental wellbeing, which contributed to the environmental and community health of the garden, which in turn deepened relationships and increased wellbeing. This is evidenced by the gardener who added a swing and benches to their garden, so other gardeners, friends, and family would feel welcome and benefit from the garden space as they did. This placemaking of the garden space both encouraged the gardeners to spend more time in the garden and extended the wellbeing benefits of the garden to others in their community.

*5.2. Agency*

Management decisions also directly influenced participant agency, which in turn impacted participant and community wellbeing. In our study, participant agency was increased by the opportunity to communicate directly with garden and network leadership, which allowed gardeners to make requests for resources, suggest changes in garden structure (e.g., plot size), or share their needs (e.g., particular kinds of seeds or tools). The gardens in our study are nonprofit-managed, hosted by the GLFGP and supported by the network-employed Refugee and Immigrant Liaison. Research shows that grassroots or community-led gardens have more voice and consistent buy-in during the gardening process than gardens that are managed top-down, which may not have as much say in decision making [4]. However, our study shows that a blend of leadership styles can effectively balance the benefits of top-down leadership, including free land use, garden and staff resources, training, and the agency of community-led initiatives. Because all garden leaders in the GLFGP network are community volunteers who either garden at the site and/or facilitate the garden for a site where they spend a lot of time, e.g., at a church, the day-to-day management in the gardens is community-led.

The reason this blend of top-down and grassroots leadership works, though, is because of the governance structure of the GLFGP. The organization supports staff to interface directly with refugee and immigrant gardeners, which facilitates voice and access to leadership for minority participants, and they actively reach out to the community about needs and goals, as evidenced by their collaboration on this project. Their example of transparency, proactivity, and honoring participant agency, which all starts with listening,

provides a model for other programs. When the gardeners in our study were made aware of the capacities and limitations of the GLFGP, they felt heard, which led to strong relationships between the gardeners and the organization.

Agency for participants in our study was also facilitated by the opportunity to do what they chose with their harvest. When gardeners donated produce to local agencies, they felt involved, and thus included, in the larger community. This capacity for donation was nurtured by in-garden and network-wide management, and it is central to the stated or implied garden purpose. Neo and Chua [39] indicate that sharing harvests is an expectation in "community-centric" gardens, which are focused on the community building aspects of community gardening above the production facility of the gardens; donation and sharing are not expectations of 'garden-centric' and not necessarily sanctioned by all community gardening initiatives.

While both 'community-centric' and 'garden-centric' gardens can foster agency through a shared commitment to purpose in the garden, this agency depends on clarity about a garden's purpose. In the absence of clarity, there can be disagreement or confusion about one's roles within the garden, and if a gardener's personal priorities conflict with the garden's focus, e.g., if a gardener seeks a community-centric experience but is participating in a garden-centric environment, then gardeners might not be confident they are contributing in meaningful ways. This can lead to a sense of exclusion rather than inclusion. Therefore, having a clear and agreed upon garden purpose, e.g., GLFGP's mission of reducing food insecurity, anchors gardeners in the particular expectations and norms of a garden's activities and their own responsibilities as members of that community [39]. This alignment of priorities increases personal wellbeing while also fostering shared responsibility to grow and care for plants collaboratively, which impacts community wellbeing. The sense of community that is cultivated through collaborative efforts, Hoffman [61] explains, facilitates community "buy in" that ultimately raises participants' sense of social connection and social capital. We observed this in our study, when gardeners voiced appreciation for the opportunity to be a part of something larger than themselves and contribute to a shared effort. While donation is tied to the 'community-centric' garden ethos [39], the literature does not specifically connect donation or sharing with ties to the beyond-garden community, which provided important openings to build relationships with colleagues and neighbors for our participants. Additionally, while immigrant and refugee growers are often seen as recipients and beneficiaries of food assistance programs [62], our research shows that they are also contributors to these systems. This participation is an act of empowerment, whereby refugees and immigrants are not simply victims of political turmoil but also actors in reshaping social relations and power dynamics in the food system [63,64]. Often, communities and organizations fail to recognize these populations as rich resources for volunteerism and giving back [64]. We found that creating spaces to honor this inclination for immigrant and refugee gardeners can be an important avenue for agency and inclusion in both the gardens and the wider community.

*5.3. Community Dynamics*

Access to physical space, guided by management decisions about membership structure, which allowed gardeners to choose to garden with other New Americans, and agency, facilitated by garden and network leadership structures, create the context for increased personal and community wellbeing. However, the interplay between individual and community wellbeing are critical for the development of social capital. The relationship dynamics within the garden can foster or stymie community building at different scales.

In our study, participants discussed the value of ethnic community building, intercultural community building, and beyond-garden community building that emerged from their garden experience. At first glance, this is an argument for interest-based gardening [40]. However, the cultural community and sense of belonging participants find in refugee and immigrant spaces can be much greater than may be observed in an interest group. It is an identity. As such, it allows for a particular kind of social engagement

and community building that is rooted in ways that shared interests alone cannot be, for one cannot simply try on new identities in the way they might take up new interests. Community building for refugees and immigrants is important in unique ways, and this nuance of identity-based gardening expands the typology of garden types to better reflect the immigrant and refugee garden experience.

**Ethnic Community**

Interviewees discussed the importance of growing and sharing culturally relevant produce and recipes within their own cultural community, especially foods that are expensive or hard to find in local stores and/or foods that are central to traditional ceremonies or gatherings. In this way, the community garden creates a "participatory landscape" where immigrants and refugees can tend to important food, medicinal, and aesthetic crops, create community with one another, and gather in celebration [60]. Growing culturally appropriate foods provides a service to the wider refugee and immigrant community, who gains access to familiar flavors and recipes, and building relationships that increase access to culturally important food, seeds, or plants provides opportunities to celebrate a shared identity [13,51,60]. Participants in our study frequently talked about the social bonds that were strengthened by sharing culturally relevant foods grown in the gardens and the respect and gratitude they received for providing those foods for others.

**Cross-cultural Community**

The garden space also brings people together in the shared identity of gardener, an identity that is separate from all other identities, including cultural identities. In our study, this shared identity led to sharing food, growing techniques, and recipes with other gardeners, which then provided opportunities to share cultural identities across difference. This sharing of knowledge related to food production and preparation is a demonstration of horticultural diplomacy, which is the use of traditional agro-ecological knowledge about plants, plant usage, and growing practices as an instrument to facilitate cross-cultural understanding and mutual understanding [65]. It is empowering for gardeners to share their knowledge and skills, and it is impactful for members of the community to learn from their peers and neighbors, which can both diversify and democratize garden education. We saw this in our gardens.

According to the concept of commensality, sharing food can also turn self-seeking individuals into a collective group [66]. In the gardens, commensality goes beyond bread and food, extending to plants, seeds, and techniques that facilitate social cohesion. Information sharing among refugee and immigrant gardeners offers an opportunity to acknowledge the value of their traditional ecological knowledge (TEK), "a cumulative body of knowledge, practice and belief, evolving by adaptive processes and handed down through generations by cultural transmission, about the relation of living beings (including humans) with one another and with their environment" [67] (p. 1252). The practice of recognizing and sharing different techniques for growing plants and preparing recipes acknowledges that immigrants and refugees bring valuable biocultural knowledge into metropolitan areas [68]. Sharing this knowledge with others is a way to reproduce and transmit TEK, while also broadening it through continued learning in place.

Often community gardens facilitate collaboration for immigrants and refugees from different cultural backgrounds, which creates an opportunity for shared purpose and connection across boundaries, perhaps even overcoming historical tensions [9]. In their study of 272 peer-reviewed publications on the socio-cultural impacts of urban agriculture, Ilieva et al. [69] identified four primary impacts of urban agriculture: (1) engaged and cohesive communities, (2) health and wellbeing, (3) economic opportunities, and (4) education. While our study reflects these findings in a broad sense, there is little discussion in the previous research that describes the extent to which gardens nurture diversity and cultural identity, which were both important drivers of inclusion and increased wellbeing for our participants. Our research contributes to this limited research on the ways that gardens can nurture cross-cultural dynamics and shows how important it is for garden leaders to create opportunities for exchange and informal dialogue across stakeholder groups.

**Beyond-garden Community**

The broadest scope of community that our participants discussed included relationships with people in the neighborhood and beyond not directly connected to the gardens, including coworkers, neighbors, and passersby. This finding reflects literature about the role of community gardens as "bridges" to extended communities [34,61] and the impacts of community gardening on community building and a sense of belonging through "neighborly engagement" [70]. This is not limited to the refugee and immigrant experience, as research shows community gardens generate social connectedness across groups [14]. However, these impacts are likely more profound for immigrant and refugee gardeners, for whom the experience of being 'seen' and acknowledged is a critical part of developing a sense of place and belonging in a new landscape.

These relationships are further facilitated by perceived safety in the garden and urban spaces. Several participants in our study explicitly mentioned the physical and emotional safety of the gardening experience. While safety is an understudied concept in community gardening literature [69], urban gardens tend to be located in the "'urban fallow' (Clark, 2001), devalued, vacant spaces awaiting reinvestment during periods of economic stagnation," [71]. These tend to be spaces not well cared for, and safety was certainly an important concern for our participants. The tenuous nature of urban gardens, often developed on land that is in transition with short-term leases [72], can lead to the question of who belongs in the city, and thus which cultural practices are welcome. Through the provision of safety measures, garden managers can demonstrate a commitment to gardener wellbeing and inclusion [73], and as such, facilitate relationships with the beyond-gardener community. This includes things like fences and lights, which contribute to a sense of physical safety for gardeners and the neighborhood more widely, as well as things like good soil and water, which contribute to emotional wellbeing in the sense that gardeners feel cared for in the garden space when they are provided the resources to do a good job. When gardeners feel included in the garden space, and when this inclusion extends to the broader urban space, they are empowered to engage with the beyond-garden community.

*5.4. Social Capital*

Our findings point to the above mechanisms of inclusion as contributors to social capital; conversely, research on community gardens shows that social capital can also enhance agency, personal wellbeing and community wellbeing [34,42]. In their qualitative study of community gardens in Melbourne, Australia, Kingsley and Townsend [14] describe social capital, i.e. "social support, connections and networking," as a primary benefit of the garden experience, though they found that these benefits remained within the garden's fences. In our study, participants indicated that these benefits extended beyond their gardens to their neighborhoods and workplaces. Alaimo et al. [74] also found that household participation in community gardening activities positively impacted neighborhood-level perceptions of social capital, which they describe as: trust and reciprocity, knowing neighbors, feeling responsible for neighborhood, and neighborhood satisfaction. In a telephone survey study (n = 1916) in Flint, Michigan, they found that "some gardens propagated neighborhood norms and beliefs, including reciprocity, helping others, neighborhood involvement, collective efficacy, sense of community, and neighborhood pride and morale" (p. 499). They continue:

> Knowing how to generate social capital may be especially valuable for distressed urban neighborhoods . . . . As they are more likely to have the needs of the community inadequately met and/or to experience neighborhood problems, such as crime and disorder. Having a household member participate in community gardening/beautification and/or neighborhood meetings was associated with more positive perceptions of bonding social capital, linking social capital, and the existence of positive neighborhood norms and values. (p. 510)

Social capital is especially important for immigrants and refugees in these neighborhoods, many of whom are new residents, which can be an isolating experience [56]. In support-

ing social capital in the garden and beyond, the community gardening experience can contribute to a sense of community, combat loneliness, and offer support in times of distress [14]. Gardeners in our study described the ways in which the garden helped them transition to Michigan by having access to outdoor space they otherwise did not have; connections within the community to acquire seeds, plants, and knowledge from others; and developing relationships within and beyond the garden boundaries. Our findings support the assertion by Harris et al. [12] that community gardens were "culturally and socially relevant place-based interventions" for the refugees in their study.

In our study, we saw evidence of all four drivers of social capital that Firth et al. [40] identified: (1) bringing people together with a shared purpose, (2) providing a meeting place, (3) creating opportunities for informal interactions like growing, cooking, and eating food, and (4) providing an opportunity to create connections with institutions and authorities, in addition to the fifth factor identified by Shostak and Guscott [36], (5) preserving cultural knowledges and practices. The physical space of the garden created the opportunity for people to come together for a shared activity, gardening, and the implied community-centric purpose of the gardens, which encouraged placemaking, agency, and donation with the broader community, provided a backbone for shared purpose and collective ownership. This created opportunities for bonding social capital. The placemaking that occurred in the gardens created opportunities for interactions with other gardeners, visitors, and beyond-the-fence neighbors, which encouraged community building. This provides evidence of bridging social capital. The sharing of gardening practices, produce, and recipes that occurred within and across ethnic communities encouraged informal interactions and connections across communities, and the ready access to garden leaders, the Refugee and Immigrant Liaison, and the network administrators, who regularly visit the gardens in the network, provided base level connections with institutions and authorities. This is an example of linking social capital. Finally, the gardeners in our study described the ways their experience in the gardens allowed them to nourish cultural knowledge and practices.

All of these opportunities and impacts are facilitated by management input across scales, and they could be further strengthened through the same channels. For example, more facilitated sharing events, garden cookbooks, potlucks and social and cultural gatherings in the gardens [60] could provide avenues for refugee and immigrant gardeners to participate in deeper and more frequent exchange of food and stories. This outcome would have meaningful impacts on gardener agency, pride, and connectivity. Seed grants or access to funding to support placemaking projects, like the addition of benches in a garden, or group workdays to manifest cultural building projects, like casitas, would also foster more informal interactions and gardener agency. Finally, few leaders in the ten gardens with high refugee and immigrant enrollment in the network at the time of our study identified as a refugee or immigrant. The only network-level leader who identified as a refugee or immigrant was the Refugee and Immigrant Liaison. Representation in leadership roles is important and would deepen the potential for linking social capital or relationships with institutions and authorities, as well as the resilience of the urban agriculture system [75].

*5.5. Limitations*

We conducted a small place-based study and more research is needed to understand inclusion for immigrant and refugee gardeners across systems. Additionally, the small size of our study means our sample only included a few participants from each ethnic group. Therefore, we cannot exclude the possibility of culturally-rooted motivations for participation or mechanisms of inclusion. Future research could identify differences across the distinct cultural groups or between the experiences of refugees and immigrants, as well as further probe barriers and challenges immigrant and refugee participants experience. Further, the nature of our interviews did not elicit much negative feedback regarding garden management or community dynamics. Thus, more research on barriers to participation would be useful to understand factors of exclusion in addition to mechanisms of inclusion.

Finally, the GLFGP is in many ways an exceptional example of a community garden network that aims to support low-income, minority, and especially refugee and immigrant gardeners. Not only do they provide free land, water, and garden resources, they host tool sharing, offer free starts and seeds, and support a full-time Refugee and Immigrant Liaison to work with the community. Other programs could learn from these examples; it would also be useful to explore specific mechanisms of inclusion in other similar communities who host different kinds of programs and garden experiences.

## 6. Conclusions

Management decisions are the input upon which the factors of inclusion we identified hinge. The output from these factors is social capital. Our research details the path through which this can occur. This is not to say that all management is good management. As Egerer and Fairbarin [76] explain, there are scenarios where gardens can become overly-managed, which can strangle gardener agency, as well as under-management, which can over-burden participants and reduce the joy of growing food in place. Navigating this balance between thoughtful management, gardener agency, and wellbeing impacts should be a goal of community gardens and networks that strive to be inclusive places for all groups, including refugee and immigrant gardeners. Therefore, especially in ethnically diverse gardens, attention should be paid to perceptions of garden structure, resource management, and garden aesthetics across groups [3,76].

Likely the most salient takeaway from our study is that being listened to by garden and network leaders, a management strategy that facilitated individual wellbeing, agency and ownership of the garden space, and community wellbeing, was a primary driver of inclusion. The patterns identified across our participants' experiences provide insight into specific ways garden programs can expand their intervention strategies to create and nurture opportunities and voice for participants from diverse cultural backgrounds. Some of the practical takeaways for in-garden dynamics to support personal and community wellbeing include the importance of centering community building across-scale, including ethnic, garden, and beyond-garden communities. This stratified definition of community and the benefits each layer brings should be a focus of community garden management and support agencies.

**Author Contributions:** Conceptualization, L.G. and V.G.P.; methodology, L.G., L.R., and V.G.P.; validation, L.G., L.R., and V.G.P.; formal analysis, L.G., L.R., and V.G.P.; investigation, V.G.P. and L.G.; data curation, V.G.P.; writing—original draft preparation, L.G., L.R., and V.G.P.; writing—review and editing, L.G. and A.H.; visualization, A.H. and L.G.; supervision, L.G.; funding, L.G.; Project administration, L.G. and V.G.P. All authors have read and agreed to the published version of the manuscript.

**Funding:** This research was supported by grants from the Michigan State University Science and Society at State (S3) program and the Michigan State University Food@State initiative.

**Acknowledgments:** The authors would like to acknowledge the partnership and labor of the Greater Lansing Food Bank's Garden Project and the generous contributions of the participants.

**Conflicts of Interest:** The authors declare no conflict of interest.

## Appendix A

**Table A1.** Interview Protocol.

| Date | Interview Questions | Objectives |
|---|---|---|
| **June** | What are your gardening practices?<br><br>• Do you use chemical controls (herbicides, pesticides)<br>• Do you use non-chemical controls (hoeing, rototilling, hand weeding)<br>• Do you use compost?<br>• Do you use mulch?<br>• Do you use soil amendments?<br>• How has the way you garden changed since you first started gardening in Lansing?<br>• How has the way you garden changed since you first started gardening in this community garden? | Gardening practices |
| **July** | How did you decide what to grow this year?<br><br>• How do you decide how much food to grow?<br>• Where do you learn about new gardening techniques?<br>• How do you find the seeds/plants to grow?<br>• If you have gardening questions, who do you ask for help?<br>• Who do you spend spend time with at the community garden?<br>• Have you created friendships here? What do you talk about?<br>• What plants do you grow that you share with (or grow for) other people?<br>• What plants do other people share with you? | Motivations, Information and produce sharing |
| **August** | Why did you join the garden?<br><br>• Why is being part of the garden important for you?<br>• Do you feel included in your garden? What makes you feel that way?<br>• Does gardening make you feel more connected to Lansing?<br>• What are the assets/benefits that you bring to your community garden?<br>• What are the benefits that gardening brings to you, your family, or your community?<br>• Do you think the plants you grow provide any benefits for this community garden? | Motivations, Inclusion |

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
