# Peer review of "Growing Community: Factors of Inclusion for Refugee and Immigrant Urban Gardeners"

_land, doi:10.3390/land12010068_

Round 1

Reviewer 1 Report

It is a very interesting topic, and for sure  attention has to be put on it. Scientifically, the paper needs improvement. There are many 'older' references used, while publications around urban gardening has flourished very well over the last 5-10 years. So these should be more included, and to be linked to recent discussions. Also the way the focus on the research, including the distinguished elements (categories? factors?) is presented, and how the cases are selected and described needs more consideration and consistency in description. In the discussion section many observations were presented, but are these based on the analysis of the cases, or mere general important issues? The link with your analysis material should be better visible, it only starts a bit at page 15. Maybe add some (anonymous) statements to your discussion that support these findings? I’ve added several comments and suggestions in the text for your benefit.

Author Response

Please see the attached table with response to each revision comment. 

Reviewer 2 Report

Dear authors,

the paper seems well written with extensive description of the research context and comprehensive interpretation of interview data. I have only few minor comments:

Comment 1: lines 188-230

There are several square brackets, which I guess need to be filled with actual names of eg. a 'pilot' city. There are several more cases like this.

Comment 2: line 221

You might want to mention that this is a product of convenience sampling.

Author Response

Please see table with response to each revision comment. 

Round 2

Reviewer 1 Report

Dear authors,

You have made major improvements to the text, my compliments. It has now a better structure, and is supported by clear, more logical and more extensive arguments. Also your rebuttal was clear and comprehensive. Congratulations with this very nice end result. Looking forward to a follow-up.

I encountered only a few minor typo’s in de PDF that was offered to me.

Line 33: “ardeners”?? Should this be gardeners?

Line 53: “Literature”?? To be added?

Line 271-285: something went wrong in the PDF; table 3 is within the text.

Line 289: should that not be chapter 4? In 297 it is okay (4.1)

Line 450: should this not be chapter 5?

Line 700: 5.4??

Line 788: chapter 6?

Author Response

Hello,

Thank you for your close reading. We've made all of these changes. We appreciate the support for the work we did on the revision. 

Best,

Angel Hammon, on behalf of the authors